# SPICEs: Survey Papers as Interactive Cheat-sheet Embeddings

**Vinay Uday Prabhu**[*]
UnifyID
vinay@unify.id

**Matthew McAteer** [*]
Deep Cell
mattzero@deepcellbio.com

**Ryan Teehan**[*]
Charles River Analytics
rsteehan@gmail.com

## Abstract

Papers are hard to write. Survey papers are just that much harder. From the authors' perspective, challenges include the responsibility to not erase out important work being done by (sometimes) adversarially aligned research groups, finding the right semantic clustering to sub-categorize individual contributions, controlling for the verbosity and length of the final paper, ensuring an optimal mixing of personal opinion and the innate narratives in the paper(s) being cited, version controlling, ease of updating, and also the aesthetics of presentation. From the reader's viewpoint, challenges include ease of reading, single-snapshot summarizability, portability, and being given the agency to edit or fork their own copies. Taking cues from the emergence of the cheat-sheet culture in machine learning and the virtues of living editable documentation and version control, we propose an interactive and live SVG format based methodology that we term SPICE: Survey Papers as Interactive Cheat-sheet Embedding. We cover the technical details behind constructing SPICEs and present an example gallery covering 'hot button' areas in machine learning such as Out of distribution detection, the 'All you need' histrionics and Transformer architectures. We have open-sourced all of the code with regards to this project here: https://anonymous.4open.science/r/7c5ee736-c876-4b90-97bd-49870eb6b63f/.

## 1 Introduction

We, the denizens of this machine learning world, are all grasping at straws desperately trying to gracefully drink from the fire-hose of publications that flood the arXiv servers every day. The intense democratization of all facets of machine learning research: papers, code, teaching materials and the computing resources (to an extent), has meant that the state-of-the-art is evolving at break-neck speeds and hence, it is natural that the literature landscape might seem chaotic and cacophonous at times. It is in the backdrop of this observation that we'd like to contextualize and celebrate the utility of well-authored survey papers.

Besides the obvious good of providing a comprehensive bird's-eye view of the field, they serve five other important roles that are often ignored:

1. These are high quality invitation notes to researchers from different domains to contribute.

2. They serve as collections of important open problems waiting to be solved.

3. They are immensely helpful as source materials that enable faster, better, and up-to-date teaching course design.

4. They are instrumental in setting the agenda for the research directions in the near future.

5. They help ease the burden of lengthy citation lists, especially for short communication papers.

---

[*]equal contribution

# A survey paper author

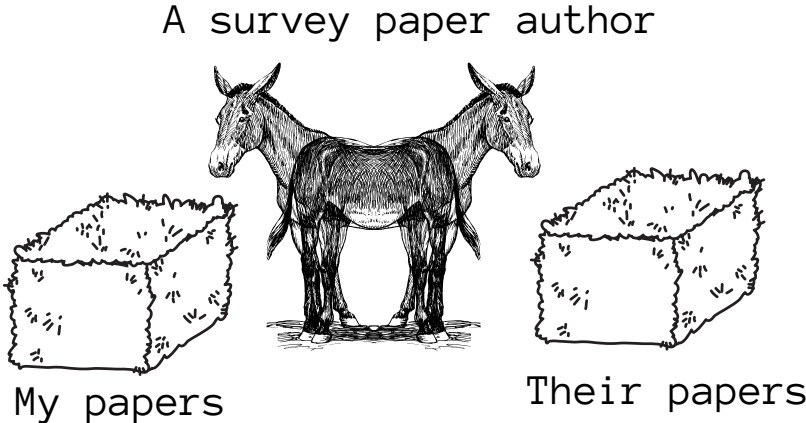

My papers              Their papers

Figure 1: Visualization of the Burdian's ass conundrum from the perspective of a survey paper author

## 1.1 THE CHALLENGES AND A NEW DISRUPTIVE-MEDIUM

As one might fathom, it is not easy to author and publish survey papers. To begin with, it is natural for a researcher to have a stronger leaning towards working on new research rather than comprehensively summarizing the legacy work in the field. While the allure of getting to buff up one's citation-counts (and $h$-index-like metrics) does act of a counter-balancing factor, the sheer volume of labor involved and the unacceptability of such work in mainstream conference venues such as NeurIPS, ICML, ICLR etc. acts as a strong detraction point. Also, we opine that gate-keeping constraints such as the demand for a co-author to be a de facto "expert"[1]) only makes things worse.

Survey papers also elicit a uniquely specific challenge: The challenge of toeing the proverbial pareto-optimal walk between one's own contributions to the field and the contribution of the other researchers and research groups (that are in some cases, adversarially situated). This necessitates imbibing a subtle personal tonality on the research ideas being explored whilst being respectful and considerate towards the work of others, an art-form in itself. The authors would like to unabashedly admit that this Buridan's ass-like quandary Weintraub (2012) has killed many of survey paper-writing efforts that they themselves initiated but never finished (See Figure 1.1). Besides this, there is also the challenge to stitch together a coherent narrative to ensure that the paper does not have the feel of a hastily copy-pasted cacophonous mix of all the paper abstracts available in that area. This sometimes does lead to what we term as the "monolithic-blob effect" in papers where all the different works covered are propelled into a single lengthy section of the paper as sub-sections without an over-arching narrative that helps categorize and contextualize the sub-sets of ideas that resulted in the constituent papers. As example, we'd like to draw the attention of the reader towards Figure 5 from *Efficient Transformers: A Survey* Tay et al. (2020) (which we found to be otherwise extremely well-authored), where oddly, a certain *Section 3.2* contained a stock-pile of all the efficient Transformer models that had been proposed in recent literature.

Lastly, there is the need to ensure that the focus is not only to include the hit-papers emanating from the usual-suspect corners of the industry and academia (the FAANG -US news-Top-$K$-schools nexus and the other "Grey-hoodie miscreants" Abdalla & Abdalla (2020) ). This responsibility also takes specific importance in the context of being a vanguard against citation erasure that elicits paying strong heed to the "Citational Desires" Murphy (2020) of contributors such as black women scholars whose contributions have been systematically under-emphasized or erased[2].

---

[1] *"The list of authors should contain at least one author who has made significant contributions to the topic over the last few years and can thus be considered an expert on that topic"*. *Source:* `https://ijcai-21.org/call-for-survey-track-papers`

[2] `https://www.citeblackwomencollective.org/`

### 1.1.1 THE RISE AND RISE OF CHEAT-SHEET COMPENDIA

"*First they ignore you, then they laugh at you, then they fight you, then you win*". - **Mahatma Gandhi**.

With these challenges in mind, the increasingly prose-averse machine learning community seems to have gravitated towards a rather intriguing hack in the form of cheat-sheets. The cheat-sheet titled *A mostly complete chart of neural networks van Veen (September); Leijnen & Veen (2020)* trended on every social-media platform upon it's release in September 2016[3]. Similarly, one of the more popular pages on a certain popular online knowledge portal[4] is literally titled *Top 28 Cheat Sheets for Machine Learning, Data Science, Probability, SQL and Big Data*. The GitHub repository, `awesome-machine-learning` has over 49,200 stars and 12000 forks [5]. The *Awesome Pytorch list*[6] has 11.2k stars and 2.5k forks. Stanford's *CS 229 - Machine Learning tips and tricks cheat-sheet* is similarly well known and has also been translated into dozens of languages[7]. Robbie Allen's compendium titled *Cheat Sheet of Machine Learning and Python (and Math) Cheat Sheets* [8] attracted 7.3K "claps"! While it is easy to don the cloak of academic snobbery and pooh-pooh away these developments as "shortcut-hacks", we implore the academic priesthood of the high "cloisters" to pay heed to this development from the humble denizens of the "bazaar" (See *The cathedral and the bazaar* Raymond (1999) ). Belonging squarely to the "bazaar" camp, we champion the cause of this cheat-sheet phenomenon, and propose a template versioning of the cheat-sheets that we term SPICEs and motivate its utility in the context of authoring and sharing survey papers.

The rest of the paper is organized as follows. In Section 2, we cover the related work from the viewpoint of the *WISE* framework. In Section 3, we present the procedure to construct the SVGs. In Section 4, we present the SPICE gallery with some examples and conclude the paper in Section 5.

## 2 A BRIEF OVERVIEW OF RELATED WORK

There have been several high profile calls for revisiting the current publication model targeting both the authorship-and-reviewing facets as well the formatting facet.

**Authorship-and-reviewing facets**: With specific emphasis on challenging the review-process and the publish-or-perish culture, Parnas' 2007 call to "Stop the numbers game" Parnas (2007) begins with the stark warning that "counting papers slows the rate of scientific progress". Yoshua Bengio's more recent monologue *Time to rethink the publication process in machine learning* Bengio (2020) imbibes a call to embrace the "Slow science manifesto" and claimed that *"... the race to put out more papers is humanly crushing"* while advocating for the journal-conference hybrid model.

**Formatting facets**: One of the central gripes of *anti-pdf-paper-ness* lies the non-intersecting worlds of static text and dynamic software. In experimentation-heavy fields such as Deep learning, this can be incredibly frustrating. With the goal to bridge the worlds of interactive data, dynamic software and the static text, Mathematica notebooks[9] emerged on the scene, fueling works such as *A New Kind of Science* Wolfram (2002). Seeking to break-free from the proprietary constraints of Mathematica notebooks, the IPython notebook revolution unfolded Shen (2014). It is indeed noteworthy that the dissemination of the first ever confirmed detection of gravitational waves was made with supplemental IPython notebook[10]. In 2014, the IPython-notebook morphed into the *Project Jupyter*Kluyver et al. (2016) that now supports over 100 programming languages and has an entire ecosystem of support built around it to auto-generate PDFs ( The notebook-as-pdf package) `https://pypi.org/project/notebook-as-pdf/` or blog-posts (Ex: Jupyter-to-medium `https://pypi.org/project/jupyter-to-medium/` or Fastpages

---

[3]`https://www.reddit.com/r/MachineLearning/comments/52q6nv/the_neural_network_zoo/`

[4]`https://www.analyticsvidhya.com/blog/2017/02/top-28-cheat-sheets-for-machine-learning-data-science-probability-sql-big-data/`

[5]`https://github.com/josephmisiti/awesome-machine-learning`

[6]`https://github.com/bharathgs/Awesome-pytorch-list#tutorials-books--examples`

[7]`https://github.com/shervinea/cheatsheet-translation`

[8]`https://medium.com/machine-learning-in-practice/cheat-sheet-of-machine-learning-and-python-and-math-cheat-sheets-a4afe4e791b6`

[9]`https://www.wolfram.com/technologies/nb/`

[10]`https://www.gw-openscience.org/tutorials/`

`https://fastpages.fast.ai/`). We'd like to encourage the reader to sift through the article on *The science paper is obsolete* Somers (2018) for a more fine-grained journey of this timeline.

Recently, Stanford hosted an entire workshop titled *Scientific Publication Beyond the Text: Sharing Research Objects* [11] that showcased non-paper dissemination such as live-sites, ORBIS -The Stanford Geospatial Network Model of the Roman World[12], PanGeo Hamman et al. (2018), Binder Forde et al. (2018), Registered Reports Chambers (2013) and OpenfMRI[13] emanating from different disciplines such as Geo-data analysis, psychology and the Life sciences. (The associated Video-playlist can be found here[14]).

With regards to the machine learning community, we'd like to highlight the following important works. Firstly, we have the Codalab Worksheets initiative[15] designed to "Run reproducible experiments and create executable papers". Secondly, given the "metrics chasing" fetish that has besieged the field, Papers-with-code[16] emerges as an invaluable community resource that hosts "Machine Learning papers, code and evaluation tables" and helps keep track of what is SoTA and what is not in the top-k accuracy hunting rat-race. Thirdly, with the emergence of document2vec frameworks and semanticity-preserving dimensionality reduction algorithms such as tSNE and UMAP, we have seen interactive web-pages providing a bird's-eye view of conference proceedings come to fruition. A brilliant example of this can be seen in Fig A which is from the ICLR paper explorer Rush & Strobelt (2020) project (See Fig A for example). Lastly, we'd like to mention other efforts such as *DLPaper2Code* Sethi et al. (2018), *Multimodal Knowledge Graph for Deep Learning Papers-and-Code* Akrotirianakis et al. (2018) and *SOLE* Pham et al. (2012) that offered a novel solution for linking research papers with science objects. Lastly, the web-page of Rethinking ML Papers has a survey of other workflows such as Distill.pub Odena et al. (2016) and Hypothesis[17]

## 3 METHODOLOGY

Inspired by the above body of work, we formulated a simple Google-slides-SVG export methodology whose details we present in this section. The rest of this section describes a workflow used to generate the eventual deliverable: A hosted hyperlinked SVG. Where possible, we also present alternatives to some steps.

### 3.1 DESIGNING THE DIAGRAM

The taxonomy does not need to be designed from the start in Google Slides. The information may originate in Google Slides [18], or it may be copied from a draft. The draft may be designed on paper, or in another program like Roam Research [19], draw.io[20], or yEd[21]. Ultimately, the information that the final SVG retains will need to be present in the Google Slides slide.

### 3.2 GOOGLE SLIDES (OR OTHER PRESENTATION PROGRAMS)

Once the diagram is laid out, it can be assembled in Google Slides. Other programs with similar functionality (e.g., Microsoft PowerPoint, Apple's Keynote) can be useful as a substitute for this step, though this workflow has been verified using Google Slides. Google slides has the advantage of being shareable among researchers working remotely, regardless of local operating systems or

---

[11]`https://psychology.stanford.edu/events/scientific-publication-beyond-text-sharing-research-objects`

[12]`https://orbis.stanford.edu/`

[13]`https://reproducibility.stanford.edu/`

[14]`https://bit.ly/3t6I0WQ`

[15]`https://worksheets.codalab.org/`

[16]`https://paperswithcode.com/`

[17]See `https://rethinkingmlpapers.github.io/submit/`

[18]`https://www.google.com/slides/about/`

[19]`https://roamresearch.com/`

[20]`https://draw.io/`

[21]`https://www.yworks.com/products/yed`

even browsers. While it is true that other design programs like Figma [22] also share these properties, we believe most engineers not typically involved in front-end engineering will be more familiar with Google Slides.

For creating a hyperlinked diagram in Google slides one can use either a default diagram (e.g., through selecting `Insert > Diagram` in the top menu) or assemble it from scratch using shapes, lines, images, or text boxes (all of which can be selected from the Insert section of the top menu, or from the menu below).

When creating diagrams in Google slides, there are a few additional features of the UI to be mindful of. 1)

### 3.3  APPLICABLE DESIGN PRINCIPLES

Before exporting the slide to SVG format, we recommend a few design principles making the output both navigable and aesthetically pleasing. These have been selected both by practical value, as well as their extolling elsewhere by designers such as Lima (2019).

**Give every design choice a purpose**    To communicate you must reduce cognitive load on the part of the viewer and emphasize what matters. Every action, color, and visual element should support data insights and understanding. Ideally, one should maintain a consistent style for conveying types of information (even better if you can provide a legend or key in your diagram).

At a high level, consider whether you want the diagram to convey to the user a hierarchy, data orientation, schema, or relationships. Even with the SVG being hosted digitally, minimize the ink one would use if they had to print it out (i.e., maximize the data-ink ratio). When creating shapes or the diagram, keep in mind that any feature of the shape can be used to convey information or lack thereof. This extends to labels, highlights, line connections, groupings of multiple shapes, or even the exact measurements of the shapes (it's good to make a habit of copying and pasting text and shapes if you want to conserve information among data subsets).

The same principles for designing a consistent map legend apply here, including the design of inconsistent parts. If you want to emphasize some part of the diagram with a design change, the strength of the emphasis will be inversely proportional to how common that design style is in the rest of the diagram.

Since the motivation behind using an SVG is partly the use of hyperlinks, one shouldn't neglect the style of the links. When adding hyperlinks to text in the diagram, you can also remove the default underline and change the default color. That said, make sure that the hyperlinked text follows a consistent color/font so that viewers can recognize where to point.

**Make use of color-coding while compensating for its limitations**    Color-coding parts of the diagram, either for selective emphasis or as part of a consistent labelling schema, can be effective. However, one should not rely solely on color changes as it has its limitations.

The first major limitations is the number of possible contrasting color combinations. Anyone that's created a scatter-plot knows the challenge of plotting more data categories than there are easily discernible colors. Richardson et al. (2014) provides some further examples of contrasting color combinations, including appropriate backgrounds. The Color Usage Research Lab at NASA Ames provides additional guidelines on color selection (`https://colorusage.arc.nasa.gov/discrim.php`).

The second major limitation is the prevalence of color vision deficiency. Large random population surveys such as Birch (2012) show that the prevalence of red-green deficiency in European Caucasians is about $8\%$ in men, and about $0.4\%$ in women and between $4\%$ and $6.5\%$ in men of Chinese and Japanese ethnicity.

If part of the intended viewing audience is color vision deficient, similar contrast effects can be achieved in greyscale. Like with the emphasis principles, keep in mind that some parts of the audience may be color vision deficient. Using features such as shape or font or contrasting shades of grey can get around this.

---

[22]`https://www.figma.com/`

**Mind where the SVG is being hosted** Bear in mind the intended audience or hosting place of the final product. Whether the final SVG is going to be a standalone page or embedded within a larger website or e-book should inform limits on some design choices. For example, if the SVG is going to be embedded within a larger web page, the chosen font will not automatically scale to the text of the web page. Setting a lower limit on text size (such as `16pt` or `18pt`) can prevent viewers from getting lost.

Of course, one should be mindful that any hyperlinks will not be conserved in printed media. If the links are truly necessary and the desired printed format is necessary, we recommend redesigning the diagrams to incorporate QR codes instead of hyperlinks. URLs, v-cards, social media information, emails, SMS links, images, documents, sound files, and app stores can all be embedded in a QR code. Multiple tools exist for generating QR codes[23]. Displaying standalone urls in a diagram in lieu of QR codes is acceptable, as long as the url is both short and memorable. QR codes are preferable if short domain names haven't been secured or if there is an abundance of links.

## 3.4 EXPORTING THE SVG

When you no longer want to make changes to your diagram, you can export the SVG. If you're using Google Slides, go to `File > Download > Scalable Vector Graphics (.svg, current slide)` in the top dropdown menu. If any images are included in the Google Slide, make sure that they're included in the same directory as the SVG, wherever it is hosted.

## 3.5 HOSTING THE SVG

Once the SVG is exported, it can be hosted on a web page or blog, or even embedded in a PDF paper.

**Basic, Temporary Hosting** If the SVG is being for a temporary presentation, it can take up the entirety of the web page. The advantage this can be cheaply hosted with single-line HTML file, as well as a free hosting service (such as GitHub Pages). For embeding the hyperlinked SVG in HTML, one can create an object tag with a `"image/svg+xml"` type as follows:[24]:

Listing 1: Single-line index.html file for fast

```
<object type="image/svg+xml" data="SVG_filename.svg">
</object>
```

The reason for this choice is because SVG is XML based and not HTML based. You can't use a normal `<a>` or `<image>` tag outside of the SVG file itself, and instead have to include links using Xlink.

If you want to change links within the SVG file itself, anchor tags are usable (though `xlink:href` should be used in place of `href`, as defined in DeRose et al. (2000)). You can do this if you want to link to pages or images that are no longer hosted in the same directory as the SVG itself.

**Long-term, link-rot-resistant Hosting** One of the requirements of multimedia conference presentations is that the material be accessible for a long period of time. Contrary to popular belief, content posted to the internet is not eternal. Much of the internet from the 1990s and early 2000s is no longer accessible (a phenomenon sometimes referred to as link-rot, such as in works like Gomes & Silva (2006)). While tools like the Wayback Machine[25] are extremely useful, its pages are often slow to load, missing sizeable amounts of content like images, and not discoverable through many search engines. If the SVG is to be stored on a blog, or within a website created with a more advanced framework, there needs to be a way of storing it beyond the horizon of 10 years beyond which most content struggles to persist.

Given the long-term storage requirements, we propose a two-part framework for long-term hosting:

---

[23]An example web tool the authors have used in the past: `https://www.qr-code-generator.com/`

[24]See `https://matthew-mcateer.github.io/oodles-of-oods/` for an example of this temporary, SVG-as-a-webpage approach to hosting.

[25]Wayback Machine: `https://web.archive.org/`

1. Rendering the text and multimedia may be done using any technology. This may be a static site generator like Gatsby.js, the JAM stack, or a custom CMS.

2. The text and multimedia must be stored as simply as possible. The SVG format solves part of this problem, but this extends to using a tool like Markdown and Git for storing everything else (even if it's just the object tag that references the SVG file).

As long as the SVGs and associated content are stored on a long-lived, erosion-resistant host, one can use any technology they please for the front-end[26]. Most Markdown-rendering tools should be able to use the object-tag-based embedding described in Listing 1.

## 4   SPICE GALLERY

In this section, we present a gallery of examples that we constructed using the technique detailed in Section-3. In doing so, we want to highlight two issues. The first is the utility of the `svg` latex package[27] built with a goal to easily include and extract SVG pictures in LaTeX documents. The second is the breadth of the topic-space that is being surveyed and SPICEd up. To this end, we pick one example each covering : A research topic (OOD failures), A model survey (Transformer models from Tay et al. (2020)) and the wild garden of *X is all you need* papers that have emerged in past few years.

To begin with, in Figure 2, we address our presentation addressing the previously highlighted stock-piling of all the efficient Transformer architectures in Tay et al. (2020) by means of a SPICEy-rendition of the same. Secondly, in Figure 3, we present a landscape of the ideas that populate the Out-of-distribution-detection corner of the machine learning world. The success enjoyed by the *Attention is all you need* paper Vaswani et al. (2017) spurred an avalanche of papers whose title or content banked off of the *X is all you need* quip. In Figure 6, we present a SPICEd version of the survey of all these papers that allows the reader to summarize where and why these quips occur.

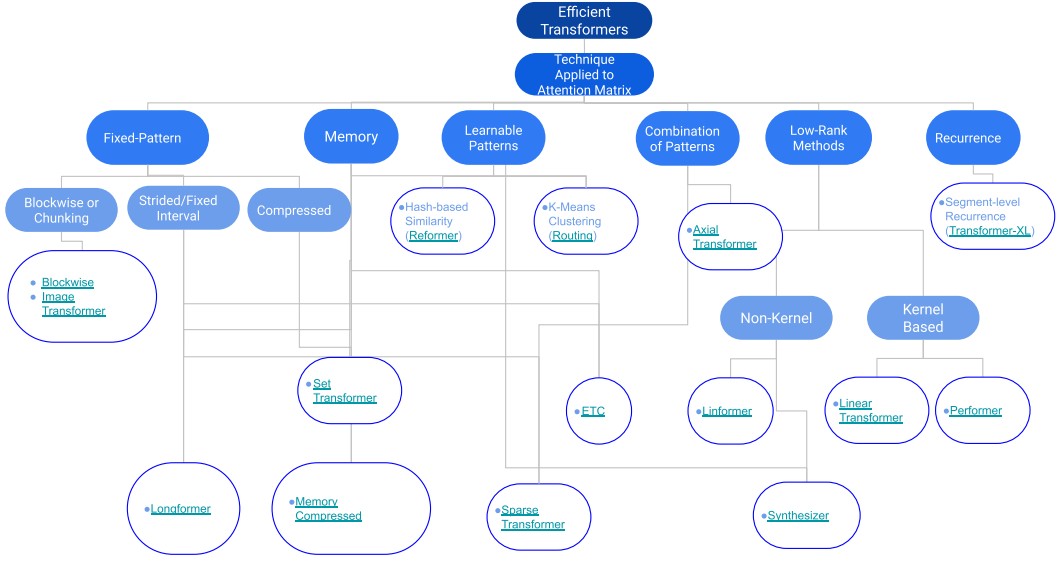

Figure 2: SPICE rendition of the survey in Tay et al. (2020)

---

[26]A example of this framework, with a hyperlinked SVG rendered with Gatsby.js, can be found here: `https://matthewmcateer.me/posts/ood-taxonomy/`

[27]`https://ctan.org/pkg/svg?lang=en`

## 5 Conclusions and future work

The institution of the scientific paper has received a lot of challenges in recent times, especially in high flux research areas such as machine learning. Researchers and community contributors alike are cleverly harnessing non-traditional media such as Twitter handles[28],Instagram[29], podcasts, YouTube, Twitch streams, and apps such as Clubhouse and TikTok, to get their message across. We feel it is inevitable that there will soon be an infusion of NFT(Non-Fungible Token)-like ideas thrown into the mix, especially given flag-posting or anti-scooping motivations.

Survey papers, specifically, are facing a unique set of challenges. To begin with, they straddle this grey-zone between knowledge and wisdom in the DIKW pyramid Frické (2019). Too much personal opinion, and you have a parochial preachy wisdom-heavy rendering that might be a disservice for someone seeking a broad-based lay-of-the-land perspective of the topic being surveyed. Similarly, mere copy-pasting and google-scholar-parsing can turn it into just a citation-mining knowledge-consolidation device. Inspired by the emergence of cheat-sheets, we present a framework that we term SPICE (Survey Papers as Interactive Cheat-sheet embedding). In this paper, we have covered the technical know-how of quickly creating SPICEs and also presented an example *SPICE-gallery*. This is a work in progress and we anticipate refining and improving this framework in multiple ways based on community feedback.

## 6 The Whyness of the SVG: An accessibility statement

As stated in Gardner & Bulatov (2008), the inclusion of *"title and description attributes in the SVG specification along with other features promoting accessibility make SVG a nearly ideal language for creating accessible graphics"*. Also, the SVG format has a slew of accessibility features with regards to sonification, braille displays and haptic feedback screens built in that made it an attractive proposition for us to adopt. The W3C Note 7( Charles McCathieNevile (2000)) specifically states: *"SVG offers a number of features to make graphics on the Web more accessible than is currently possible, to a wider group of users. Users who benefit include users with low vision, color vision deficient or sight impaired users, and users of assistive technologies."*. Further, we are using the guidelines from the **Tree of graphic** section in the "SVG Accessibility/People and Issues" section[30] and formatting the surveys using the following order of : title, axes, legends, data and reference elements adhering to the National Center for Accessible Media (NCAM) suggestions. Many of these recommendations were taken into consideration during our design process (See section-3 3.3). Further, with regards to our initial draft, we have received feedback on adding aural tones to the individual data elements (so the reader can hear when they mouse over the data element) which we plan to implement during our demo. Last but not the least, we stand committed to enacting the changes that the reviewers and the workshop participants might suggest.

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

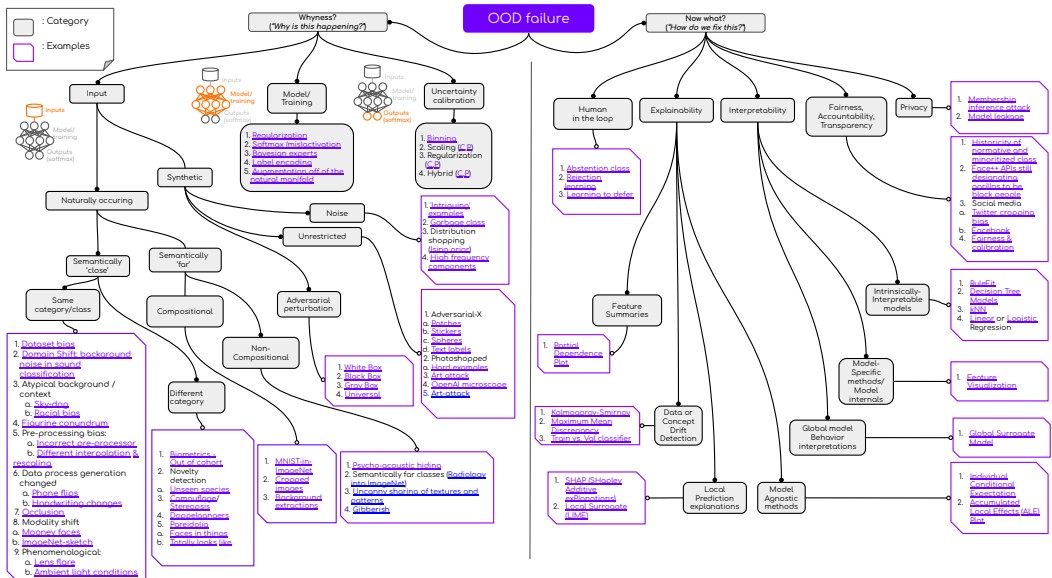

Figure 3: SPICE rendition of the field of *Out of Distribution* failures

Marja-Riitta Koivunen Charles McCathieNevile. Accessibility features of svg. `https://www.w3.org/TR/2000/NOTE-SVG-access-20000807/`, Aug 2000. (Accessed on 03/16/2021).

Steve DeRose, Eve Maler, David Orchard, and Ben Trafford. Xml linking language (xlink). *W3C Working Draft*, 19, 2000.

Jessica Forde, Matthias Bussonnier, Félix-Antoine Fortin, Brian Granger, Tim Head, Chris Holdgraf, Paul Ivanov, Kyle Kelley, M Pacer, Yuvi Panda, et al. Reproducing machine learning research on binder. 2018.

Martin Frické. The knowledge pyramid: the dikw hierarchy. *KO KNOWLEDGE ORGANIZATION*, 46(1):33–46, 2019.

John A Gardner and Vladimir Bulatov. Making scientific graphics accessible with view-plus iveo. In *Proceedings of The 23rd Annual International Technology & Persons with Disabilities Conference. Los Angeles, CA: CSUN*, 2008.

Daniel Gomes and Mário J Silva. Modelling information persistence on the web. In *Proceedings of the 6th international conference on Web engineering*, pp. 193–200, 2006.

Joseph Hamman, Ryan Abernathey, C Holdgraph, Yuvi Panda, and Matthew Rocklin. Pangeo and binder: Scalable, shareable and reproducible scientific computing environments for the geosciences. In *AGU Fall Meeting Abstracts*, volume 2018, pp. IN53A–03, 2018.

Thomas Kluyver, Benjamin Ragan-Kelley, Fernando Pérez, Brian E Granger, Matthias Bussonnier, Jonathan Frederic, Kyle Kelley, Jessica B Hamrick, Jason Grout, Sylvain Corlay, et al. *Jupyter Notebooks-a publishing format for reproducible computational workflows.*, volume 2016. 2016.

Stefan Leijnen and Fjodor van Veen. The neural network zoo. In *Multidisciplinary Digital Publishing Institute Proceedings*, volume 47, pp. 9, 2020.

Manuel Lima. Design — communication — data visualization. *Material Design*, 2019. URL `https://material.io/design/communication/data-visualization.html`.

Hannah Murphy. Nash calls for 'stewardship' in black feminist citation — the heights. `https://www.bcheights.com/2019/11/17/nash-calls-for-stewardship-in-black-feminist-citation/`, June 2020. (Accessed on 03/09/2021).

Augustus Odena, Vincent Dumoulin, and Chris Olah. Deconvolution and checkerboard artifacts. *Distill*, 1(10):e3, 2016.

David Lorge Parnas. Stop the numbers game. *Communications of the ACM*, 50(11):19–21, 2007.

Quan Pham, Tanu Malik, Ian Foster, Roberto Di Lauro, and Raffaele Montella. Sole: linking research papers with science objects. In *International Provenance and Annotation Workshop*, pp. 203–208. Springer, 2012.

Eric Raymond. The cathedral and the bazaar. *Knowledge, Technology & Policy*, 12(3):23–49, 1999.

Rick T Richardson, Tara L Drexler, and Donna M Delparte. Color and contrast in e-learning design: A review of the literature and recommendations for instructional designers and web developers. *MERLOT Journal of Online Learning and Teaching*, 10(4):657–670, 2014.

Alexander M Rush and Hendrik Strobelt. Miniconf–a virtual conference framework. *arXiv preprint arXiv:2007.12238*, 2020.

Akshay Sethi, Anush Sankaran, Naveen Panwar, Shreya Khare, and Senthil Mani. Dlpaper2code: Auto-generation of code from deep learning research papers. In *Proceedings of the AAAI Conference on Artificial Intelligence*, volume 32, 2018.

Helen Shen. Interactive notebooks: Sharing the code. *Nature News*, 515(7525):151, 2014.

James Somers. The scientific paper is obsolete. here's what's next. `https://www.theatlantic.com/science/archive/2018/04/the-scientific-paper-is-obsolete/556676/`, April 2018. (Accessed on 03/10/2021).

Yi Tay, Mostafa Dehghani, Dara Bahri, and Donald Metzler. Efficient transformers: A survey. *arXiv preprint arXiv:2009.06732*, 2020.

Fjodor van Veen. The neural network zoo - the asimov institute. `https://www.asimovinstitute.org/neural-network-zoo/`, 2019 September. (Accessed on 03/09/2021).

Ashish Vaswani, Noam Shazeer, Niki Parmar, Jakob Uszkoreit, Llion Jones, Aidan N Gomez, Lukasz Kaiser, and Illia Polosukhin. Attention is all you need. *arXiv preprint arXiv:1706.03762*, 2017.

Ruth Weintraub. What can we learn from buridan's ass? *Canadian journal of philosophy*, 42(3-4): 281–301, 2012.

Stephen Wolfram. *A new kind of science*, volume 5. Wolfram media Champaign, IL, 2002.

## A  APPENDIX-A: PLOTS

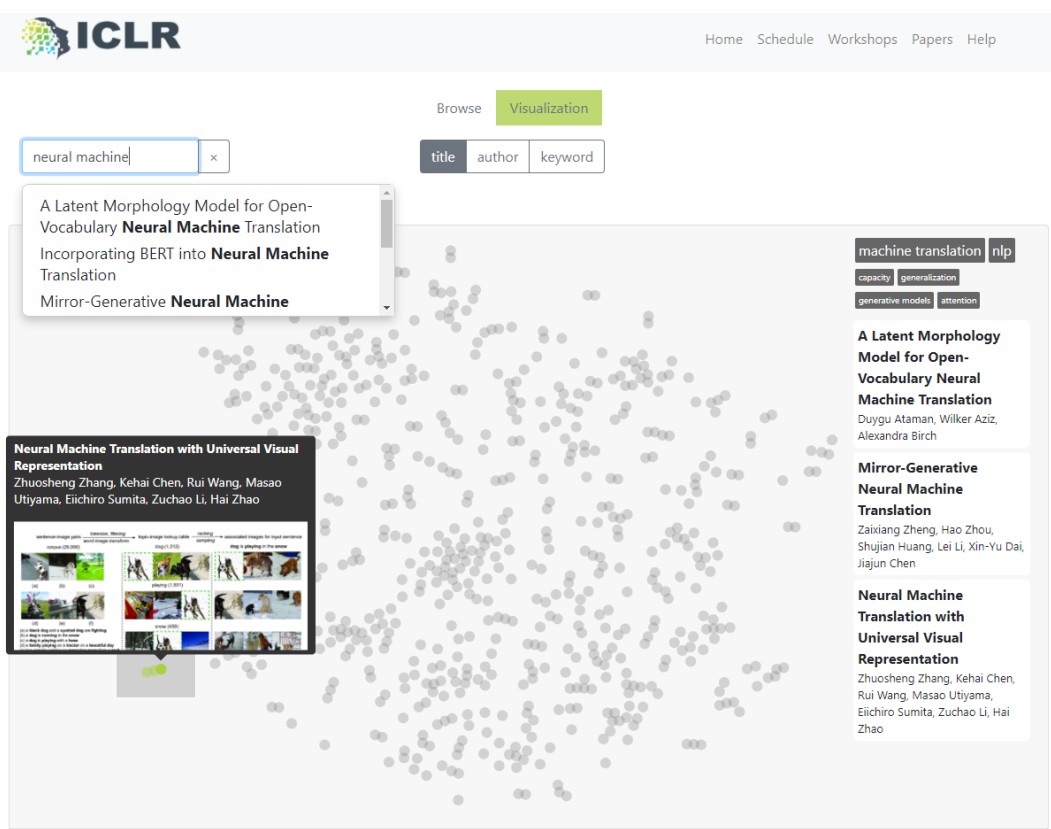

Figure 4: Visualization of article-abstracts using pretrained sentence embedding model and t-SNE 2D embeddings in the ICLR paper explorer built with the MiniConf open-source software

## Contents

Figure 5: Figure exemplifying the monolithic bloating effect addressed in Section 1.1 from Tay et al. (2020). (This specific *Table of contents* seen in Fig 5 is NOT available in the paper Tay et al. (2020) and was generated using the `\tableofcontents` latex command).

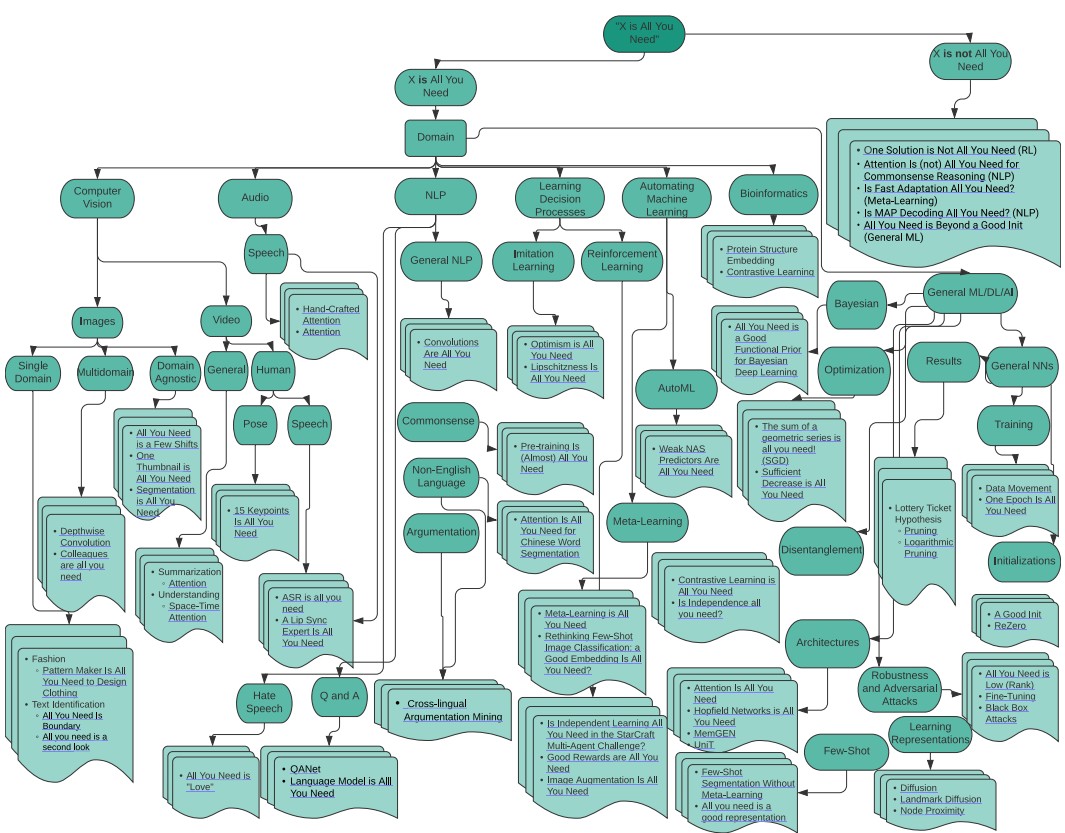

Figure 6: SPICE rendition of the glut of *X is All You Need* machine learning papers

