# OpenReview forum: "SPICES: SURVEY PAPERS AS INTERACTIVE CHEATSHEET EMBEDDINGS"
_ICLR.cc/2021/Workshop/Rethinking_ML_Papers — Rethinking ML Papers - ICLR 2021 workshop Poster_

### Official Review · AnonReviewer1 · 2021-03-28
**A joy to read, and important ideas for how to improve the survey paper**

**Accessibility:**

Score of 4 (Strong): Submission states accessibility concerns and provides solutions within the proposed framework. However, it does not declare the limitations and exceptions.

**Litreview:**

Score of 4 (Strong): The submission directly differentiates itself from previous works and formats.

**Problemstatement:**

Score of 4 (Strong): The submission sets a very strong example of how to address the problem, which should be relevant to the workshop themes.

**Relevance:**

Score of 4 (Strong): The submission directly addresses a theme of the workshop, and does so in a very professional manner.

**Results:**

Score of 4 (Strong): Submission is very well structured and follows all the criteria (i.e. clarity, novelty, interactivity, and coherency). However, practical significance/theoretical implications are not discussed.

**Reviewerconfidence:**

I am quite confident in my assessment as I understood the work well and am familiar with the related work.

**Reviewtext:**

This work proposes an alternative to the usual survey paper format in machine learning. Motivated by the difficulties in balancing breadth and depth in surveys, the submission draws inspiration from the recent prevalence of cheatsheets in ML to construct a cheatsheet-type survey paper. The proposed cheatsheet is essentially a flowchart in SVG format, with hyperlinks, logical hierarchies, colour-coding, and shape-coding. There is substantial discussion in this work on both the theoretical design and practical implementation issues (section 3), especially in regards to maintaining and improving accessibility.

I greatly enjoyed the overall writing style in the paper; it felt quite alive and befitting of the overall theme of the workshop.

While I think the work contributes valuable ideas in a clear framework--and should be accepted in my opinion--, the following should be improved for the camera-ready version.
- More explicit discussion would be nice of how the SVG cheatsheets contribute to solving the problems with survey papers that were identified in the introduction. For example, I think some discussion of how cheatsheets force a certain perspective (perhaps one can't have too narrow a focus when building such a cheatsheet) would be valuable.
- Some discussion of limitations and relation to traditional survey papers would be interesting. I wasn't clear on whether SPICES was supposed to be a complete alternative to traditional surveys, or a complement/go-between. It could be that a cheatsheet could supplement a survey paper, and provide a sort of abstract. It could also be that cheatsheets are a sort of intermediary "on the way" to a traditional survey paper.

**Score:**

Accept: The reviewer believes the submission provides a novel and reliable scheme to improve science communication but needs improvement.

---

### Official Review · AnonReviewer2 · 2021-03-31
**Well summarised "survey" of survey papers, needs more clarity on method proposed**

**Accessibility:**

Score of 5 (Exceptional): Submission identifies and articulates accessibility matters, provides justifications for the proposed paradigm, and declares the limitations.

**Groundsforrejection:**

- More clarity is needed on what exactly is expected out of the end user wanting to try this out.

- The link provided does not host any understandable content.

- The authors do not specifically pinpoint how this format is interactive.

- How does the novelty of this format compare to existing platforms/tools which may host flow charts?

- (Not a major issue) The language of this paper is set in an informal tone : which is appreciated in many places but at the same time, could be avoided to bring about more clarity.
E.g. - "Too much personal
opinion, and you have a parochial preachy wisdom-heavy rendering that might be a disservice for
someone seeking a broad-based lay-of-the-land perspective of the topic being surveyed."



**Litreview:**

Score of 3 (Neutral): The submission acknowledges previous work, but does not necessarily explain how the submission differentiates itself (i.e we want to avoid the “deluge of citation” strategy, leaving the reviewer to click through references and figure this part out for themselves).

**Problemstatement:**

Score of 4 (Strong): The submission sets a very strong example of how to address the problem, which should be relevant to the workshop themes.

**Relevance:**

Score of 3 (Neutral): Attempt was clearly made to address a theme of the workshop, but it seems that the work was ‘retrofitted’ to match the theme of the workshop.

**Results:**

Score of 2 (Needs Improvement): Submission shows a poor level of clarity, novelty, coherency, and interactivity.

**Reviewerconfidence:**

3
Would like to check if there is any issue with the link mentioned.

**Reviewtext:**

The given paper presents challenges of disseminating research in the context of survey papers - and proposes the SVG format as an alternate concise way of sharing this information. The authors are motivated by the cheat sheet culture which has gained prevalence in the fast-paced field of machine learning - where one-stop-shop resources have demonstrated wide popularity.

The authors propose a Google slides SVG export methodology. They claim that this is a “live” and “interactive” format, presumably due to the hyperlinks embedded in these SVGs. To produce these cheat sheets, the authors expect users to create diagrams in Google Slides. They summarise certain guiding design principles and encourage users to adhere to these while building the schematic diagrams.

This paper is a well summarised “survey” on survey papers.
The authors specify the challenges faced during writing review papers and drive home the point through interesting references, such as the Burdian’s ass reference. They stress on how survey papers in the form of cheat sheets are more portable and editable. The interoperability aspect has also been addressed in this paper.

It is unclear at times as to what exactly the methodology is:  whether the authors propose a tool for rendering “SPICE”d up versions of a survey paper, or expect the reader to do so manually.
The title of the paper contains "interactive" - there is no explicit emphasis on the liveness of this format.  It would have been clearer if the authors tied back the novelty of this format to the challenges initially mentioned, if this was the intention.

The solution link mentioned doesn't contain any code except for an SVG which was not reproducible on the reviewer's machine - clarification is needed on the intended content.

It would have been nice to see a comparison with similar flow chart based methods/platforms of communication. The paper does not touch upon how this format could be aptly peer-reviewed by the academic community.


**Score:**

Accept: The reviewer believes the submission provides a novel and reliable scheme to improve science communication but needs improvement.

---

### Meta-Review · Area_Chair1 · 2021-03-31

**Recommendation:** Accept
**Confidence:** 5

**Metareview:**

Pros: The papers explores the problems with constructing an adequate survey, from reviewer 2, ' "survey" on survey papers', in an interesting and thorough treatment. They include scholarship and ideas from a variety of corners, such as the need to cite a variety of sources, how to treat the papers in a balanced way, etc.  They propose utilizing a cheatsheet-style graphical format via SVGs to illustrate and/or guide the construction of a survey, or illustrate the survey for others and allow it to be editable.

There is a nice section on accessibility.

Cons: Both reviewers pointed out areas to improve the overall clarity of the work.  First, the motivation around survey papers and the difficulties with writing surveys, as well as the benefits and popularity of cheatsheets is clear in the first half of the paper, but it is not  clear about how those ideas are linked explicitly to SPICES and how the paper intends the reader to use SPICES in the second half of the paper (starting at 3: Methodology). The interactivity of the SVG format is also not very apparent in the current draft.

---

### Decision · Program_Chairs · 2021-04-01

Accept (Poster)